# Gastric Cancer in Northern Canadian Populations: A Focus on Cardia and Non-Cardia Subsites

**DOI:** 10.3390/cancers11040534

**Published:** 2019-04-15

**Authors:** Amy Colquhoun, Heather Hannah, André Corriveau, Brendan Hanley, Yan Yuan, Karen J. Goodman

**Affiliations:** 1School of Public Health, University of Alberta, Edmonton, AB T6G 1C9, Canada; yan.yuan@ualberta.ca (Y.Y.); karen.goodman@ualberta.ca (K.J.G.); 2Department of Health and Social Services, Government of the Northwest Territories, Yellowknife, NT X1A 2L9, Canada; heather_hannah@gov.nt.ca (H.H.); andre_corriveau@gov.nt.ca (A.C.); 3Department of Health and Social Services, Government of Yukon, Whitehorse, YT Y1A 1Z4, Canada; brendan.hanley@gov.yk.ca; 4Faculty of Medicine and Dentistry, University of Alberta, Edmonton, AB T6G 2E1, Canada; 5The CAN*Help* Working Group, University of Alberta, Edmonton, AB T6G 2E1, Canada; canhelp@ualberta.ca

**Keywords:** gastric cancer, cardia, incidence, epidemiology, Arctic, Canada, Indigenous health, *Helicobacter pylori*

## Abstract

In northern Canada where there is a high prevalence of *Helicobacter pylori* infection, there is a paucity of information on gastric cancer by the topographical subsites cardia (CGC) and non-cardia (NCGC). Here we describe the incidence of CGC and NCGC, separately, among northern Canadian populations. We used data from the Cancer Incidence in Five Continents Volumes X (CI5X) and XI (CI5XI) to obtain CGC and NCGC incidence for Canada and for Yukon (YT), a northern Canadian territory. Using these data with those provided by the Government of the Northwest Territories (NT), we estimated standardized incidence ratios comparing northern populations to Canada as a whole. We also estimated age-standardized incidence rates to permit comparisons across populations globally. NT and YT populations were disproportionately impacted by gastric cancer, particularly NCGC. This was especially true for Indigenous populations: NCGC incidence rates among NT Indigenous men were 2.7 times the rates among all men in Canada, while rates among NT Indigenous women were 3.1 times the rates among all women in Canada. Similarly, age-standardized rates of NCGC among Indigenous NT residents were comparable to global regions where there is a high burden of NCGC. This study has, for the first time, quantified the incidence of CGC and NCGC for the NT and YT, providing new insights into the burden of these cancers among northern Canadian populations.

## 1. Introduction

Gastric cancer is a major contributor to the global burden of cancer: in 2018, there were an estimated 1,033,701 new cases and 782,685 deaths due to gastric cancer, making it the 6th most commonly diagnosed and 3rd most fatal cancer worldwide [1]. Gastric cancer incidence varies by sex, with rates among men on average twice those observed among women. Variation also exists across geographic regions: Gastric cancer incidence rates are high in countries within Asia, Eastern Europe, and parts of Central and South America, while rates are comparatively low in many higher-resourced regions such as North America and Western Europe [1].

Although often reported as a single entity, gastric cancers can be subdivided into two topographical categories: cardia gastric cancer (CGC) arising in the proximal area of the stomach, and non-cardia gastric cancers (NCGC) arising in more distal regions. As with all gastric cancers combined, there is considerable geographic variation in incidence across these subsites [2]. This may result, in part, from differences in risk factors by subsite that vary by geography. These might include genetic risk factors: For example, genetic polymorphisms in certain DNA repair processes can contribute to an increased risk of carcinogenesis [3,4]. Environmental factors also play a role: NCGC risk is increased by *Helicobacter pylori* infection [5,6,7] and cancers of this subsite are more common in less-resourced countries [8,9]. In contrast, CGC is positively associated with obesity and gastroesophageal reflux but not with *H. pylori* infection, evidence that suggests a distinct etiology from *H. pylori*-associated NCGC. Due at least in part to its association with *H. pylori* infection, NCGC is also associated with indicators of low socioeconomic status such as household crowding, low income, low education levels and increased number of siblings [10]. The mechanisms by which genetic and environmental risk factors interact to modify gastric cancer risk remain unclear [3,4].

Lower and steadily declining incidence of NCGC among higher-resourced countries may be attributable to improved sanitation and other factors that reduced the prevalence of *H. pylori* infection, as well as other technological advances of the early 20th century that reduced the occurrence of diseases of the pre-modern era [11,12,13]; however, some populations within more developed areas are still at high risk. Studies have documented, for example, a higher burden of gastric cancer in Indigenous populations worldwide compared to their non-Indigenous counterparts [14,15]. Similarly, others have shown elevated incidence rates across the circumpolar region, particularly among Indigenous populations [16,17]. Given the disproportionate impact of *H. pylori* on NCGC incidence relative to CGC incidence, the distribution of gastric cancer by subsite varies by *H. pylori* prevalence within higher-resourced countries. Ethnic minorities in the US, such as African-Americans or Asian-Americans, have higher rates of NCGC and lower rates of CGC compared to non-Hispanic white Americans [18,19]. This differing pattern in NCGC and CGC incidence rates has also been observed among Indigenous populations compared to their country-specific non-Indigenous counterparts [14]. There remains a paucity of information, however, on differences in gastric cancer incidence by subsite among high-risk populations.

In northern Canada, where Indigenous groups comprise a high proportion of the population, communities have expressed concern over their perceived high frequency of stomach cancer [20]. To address these concerns, a community-driven research program conducted by the Canadian North *Helicobacter pylori* (CAN*Help*) Working Group brings together academic researchers, healthcare providers, health officials, and members of communities in the Northwest Territories (NT) and Yukon (YT), Canada to conduct community *H. pylori* projects. These projects have estimated a high prevalence of *H. pylori* infection in participating communities [20,21]. Furthermore, participants assessed pathologically through biopsies of the stomach lining had a high prevalence of precancerous lesions of the stomach [20]. To characterize the burden of stomach cancer in the northern regions where these communities reside, we aimed to describe the incidence of gastric cancer overall and by subsite and compare these estimates to those observed elsewhere.

## 2. Results

### 2.1. Gastric Cancer Overall

During 2003 through 2012, there was an annual average of 2638 cases of gastric cancer in Canada; in each of the two northern territories included here, YT during 2003–2012 and NT during 1997–2015, there was an annual average of 3 cases (Table 1). In all populations defined by geography or Indigenous status, there were proportionally more cases of gastric cancer observed among men than women. Disparities in the distribution of age at diagnosis were apparent by geography and Indigenous status. In Canada, less than 25% of gastric cancer cases were diagnosed in individuals under the age of 60 (Figure 1). Among northern populations, however, over 40% of cases were observed in those younger than 60 years of age: 42% of total cases in YT and the non-Indigenous NT population, and 48% in the Indigenous NT population. The disproportionate impact on younger populations was particularly pronounced among Indigenous people in the NT: 16% of cases among NT Indigenous people were observed in 0–39-year-olds (7/44) compared to 2% in Canada as a whole (646/26,382).

### 2.2. Cardia and Non-Cardia Subsites

The proportion of total gastric cancer cases classified as cardia varied by geography and Indigenous status (Figure 2). In Canada, YT, and the non-Indigenous NT population, the proportional CGC incidence ranged from 29–37% of total gastric cancer cases. In contrast, only 2% of gastric cancer cases in the Indigenous NT population were classified as CGC (1/44). This population also had the lowest proportion of cases classified as ‘other and not otherwise specified’. The distribution of gastric cancer by subsite also varied by sex. Among men, CGC cancers comprised an average across subgroups of 25% (range 4–44%) of total gastric cancer cases; among women, this proportion was 9% (range 0–25%) on average across subgroups.

### 2.3. Comparisons Across Populations

NCGC incidence rates were higher among Indigenous NT men compared to all Canadian men (SIR 2.7, 95% CI 1.7–3.9) (Table 2). Similarly, Indigenous NT women had a higher incidence of NCGC compared to all women in Canada (SIR 3.1, 95% CI 1.8–4.8). For non-Indigenous NT men and women, incidence rates were similar to Canada. However, in YT, NCGC rates among both men and women were elevated compared to Canada (30% and 50% higher, respectively). It should be noted, however, that for YT and non-Indigenous NT groups, these differences were statistically imprecise. For CGC, with the exception of YT men and women and non-Indigenous NT men in whom there was an apparent elevation, rates were lower compared to Canadian rates. However, for most groups there were no or very few cases of CGC, resulting in imprecise estimates vulnerable to small fluctuations in incidence.

Comparisons with other regions across the globe revealed similar patterns of variation in the distribution of gastric cancer subsites by geography and Indigenous status. For CGC among men, ASRs for Canada, YT, and the non-Indigenous NT population (Table 3) were comparable to regions such as Oceania, Southern Europe, Northern Africa & Western Asia (ASRs ranging from 2.5 to 3.3 per 100,000). These estimates were low relative to high-incidence regions such as Eastern/Southeastern Asia and Eastern Europe. Among Indigenous and non-Indigenous NT women, there were no cases of CGC. For Indigenous NT men, CGC incidence rates were based on a very small number of cases and, as a result, were very low. Conversely, NCGC incidence rates were markedly high in the Indigenous NT population (ASRs 13.3 and 7.7 per 100,000 among men and women, respectively), similar to high-incidence regions like Eastern Europe. For all of Canada, and in YT men and women and non-Indigenous NT men, NCGC rates were comparable to regions such as Northern and Western Europe (ASRs in Canada, YT and non-Indigenous NT groups ranged from 5.0 to 6.1 per 100,000 men and 2.8 to 4.5 per 100,000 women). The NCGC incidence rate among non-Indigenous NT women was even lower (ASR 2.0 per 100,000) (Figure 3a,b).

### 2.4. Sensitivity Analyses

As part of a sensitivity analysis to assess the impact of the attribution of all C16.8-9 (gastric cancers classified as overlapping and not-otherwise specified) cases to NCGC, we estimated SIRs for NCGC defined as C16.1-4 (Appendix A). SIR estimates based on this revised NCGC case definition were similar to those estimated using the original definition for NCGC (C16.1-9). For Indigenous people in the NT, SIR estimates increased with the more restricted case definition: In men, the SIR for NCGC increased 33.3% to 3.6 (95% CI 1.9–6.0); in women, the SIR increased 58% to 4.9 (95% CI 2.5–8.6).

Compared to ASRs estimated using the broader case definition, ASRs estimated using NCGC defined as C16.1-4 were lower for Canada and for the study populations in the NT and YT. This, in turn, resulted in lower estimates among the study groups and for Canada relative to the global populations that maintained a broader definition of NCGC (Appendix A).

## 3. Discussion

To characterize stomach cancer burden in NT and YT, we estimated the incidence of gastric cancer overall and by subsite and compared these estimates to rates estimated for Canada nationwide. We observed that NT and YT populations are disproportionately impacted by gastric cancer compared to Canada, particularly at younger ages when a larger proportion of NT and YT cases are diagnosed. This was especially true for the Indigenous NT population, in whom 16% of cases were observed in those under the age of 40, in contrast to 2% of cases in Canada as a whole diagnosed in this age group. Further, NCGC incidence rates among NT Indigenous men and women were 2.7 and 3.1 times, respectively, rates of Canada as a whole. To provide a global context for these observations, we also compared the estimated NT and YT rates to those observed elsewhere globally and found that NCGC incidence rates in the Indigenous NT population were similar to regions such as Eastern Europe where there is a high burden of gastric cancer.

Our analysis is consistent with previous work that demonstrated elevations in gastric cancer rates across northern populations. Young, et al. (2016), for example, reported that gastric cancer rates among Inuit populations in the circumpolar region of Canada are higher than global rates [16]. Similarly, other work has shown that gastric cancer rates are higher in northern regions compared to rates observed in more southern parts of the same country [22,23]. In addition to geographic variation, our findings are consistent with reports of differences in the stomach cancer burden among Indigenous populations compared to their non-Indigenous counterparts [14]. In some regions, gastric cancer rates among Indigenous peoples were over two times the rates of non-Indigenous people living in the same area [14]: in Alberta for example, a Canadian province south of NT, during 2009-2016 gastric cancer rates among First Nations men and women were 1.5 and 2.3 times, respectively, rates of their non-First Nations counterparts [24].

As with all gastric cancers combined, there is also geographic variation in the incidence of gastric cancer subsites [2]. Because *H. pylori* is a risk factor for gastric cancer arising in distal portions of the stomach [6], it is not surprising that regions with higher prevalence of this infection are impacted more greatly by NCGC. Further, elevations in NCGC incidence have been observed to coincide with an increased burden of *H. pylori* infection among Indigenous populations [25,26,27]. Comparing Indigenous ethnic groups (designated as American Indians and Alaska Natives) to non-Hispanic whites in the United States, for example, estimated NCGC incidence rates during 1999–2004 were 5.8- and 14.9-fold higher among men and women, respectively [28]. Other reports similarly estimated NCGC incidence rates to be higher among Native Americans compared to other ethnic groups in the United States [29,30] and among Maori New Zealanders compared to non-Maori New Zealanders [31]. Despite the higher burden of *H. pylori* infection and gastric cancer among Indigenous populations, there is relatively little information on CGC and NCGC in these groups across the globe.

While our analysis of gastric cancer incidence by subsite in northern Canada contributed to reducing this information gap, there were data restrictions that limited this assessment. When disaggregating rates by age category, the CI5X and CI5XI datasets combined gastric cancers of the lesser and greater curvature (C16.5 and C16.6) with those classified as ‘overlapping lesion of stomach’ and ‘stomach, not otherwise specified’ (C16.8 and C16.9). Because C16.5-6 are typically classified as cancers of the non-cardia gastric region [19,32,33], and because other analysts have reclassified all or some C16.8-9 cases as NCGC when these classifications are relatively common [2,34,35,36], we chose to attribute all cases classified as C16.5-9 to NCGC. It is possible that some of the cases classified as overlapping or unknown topography (C16.8-9) are, in fact, cancers of the cardia and were misclassified here; however, given the low proportion of cases classified as CGC in global populations, the degree of any potential misclassification is likely small.

To assess the extent to which misclassification of cases could have impacted comparisons of NCGC incidence estimates across populations, we also performed a sensitivity analysis where NCGC cases were redefined as C16.1–4: We found that SIR estimates with the more restricted definition were similar to those estimated using the original definition for NCGC (C16.1-9). In contrast, ASRs based on the restricted definition of NCGC (C16.1-4) resulted in lower estimated rates for the study groups and Canada relative to the global populations, which maintained a broader definition of NCGC. Despite this, ASRs of NCGC among Indigenous populations in the NT remained high relative to global estimates.

The relatively small number of gastric cancer cases observed in northern Canadian populations created imprecise ASR estimates that are vulnerable to large fluctuations if the numerator is off by as little as one case. To account for this, we used SIRs to compare the observed number of cases in the study populations with the number of cases expected based on Canadian rates. This method of age-adjustment is beneficial when age-specific rates are unstable due to small numbers [37]. Unfortunately, because the age distribution of the study population is used as the standard for these methods, age adjustment through the use of SIRs limits comparisons over time or across multiple populations [38]. To support comparisons with global rates, we also estimated ASRs, using CIs to acknowledge imprecision in these age-standardized rates due to small numbers of cases.

It should be noted that we used data on gastric cancer in Canada during 2003–2007 as a comparison to rates that included cases diagnosed in more recent years in NT and YT. Given that gastric cancer rates are decreasing in Canada [39], in line with observations that NCGC rates are declining in developed regions of the world [40,41,42], our SIRs comparing NT and YT rates to Canadian rates are likely to be underestimated. While gastric cancer rates are decreasing overall in developed regions of the world, Anderson, et al. (2018) demonstrated through stratification by age group that NCGC rates may be increasing in younger non-Hispanic white populations in the United States, particularly among women [33]. Although the number of cases in NT and YT populations were too small to examine age-specific incidence rates in detail, increasing incidence in younger age groups over time could explain why the proportion of gastric cancer cases corresponding to younger age groups was larger in NT and YT than in Canada as a whole, given that NT and YT data included cases diagnosed in more recent years.

This study has, for the first time, estimated the incidence of CGC and NCGC among NT and YT populations. Consistent with other work describing gastric cancer among northern and Indigenous populations, we found that NT and YT populations were disproportionately impacted by gastric cancer, particularly of the non-cardia gastric region. These estimates provide new insights into the burden of gastric cancer in northern Canadian populations. As a result, this work supports the on-going community-driven CAN*Help* Working Group research program. In particular, it will be used to inform gastric cancer control strategies aimed at achieving the goal identified by impacted Indigenous Arctic communities of implementing effective interventions to reduce their disproportionate gastric cancer disease burden.

## 4. Materials and Methods

### 4.1. Comparison Populations

The northern populations available for comparison in this analysis are the NT population dichotomized as Indigenous or non-Indigenous, and the YT population without classification by Indigenous status. As a reference population, we use the total population of Canada, including the NT and YT populations, without classification by Indigenous status. In 2016, Indigenous groups comprised approximately 51% of the NT population of 41,135, 23% of the YT population of 35,110, and 4.9% of the total Canadian population of 34,460,065 [43]. Sex-specific frequencies were available for all comparison populations. Classification by Indigenous status was available for the NT population, but not for the YT or total Canadian populations. NT residents were dichotomized as Indigenous, including individuals who identify as Inuit, First Nations, or Métis, or non-Indigenous, including everyone else.

### 4.2. Data Sources

We used data from the Cancer Incidence in Five Continents (CI5), volumes X (CI5X) and XI (CI5XI) published online by the International Agency for Research on Cancer [44,45]. These volumes are the most recent CI5 publications: They report data from select registries around the world for 2003–2007 and 2008–2012, respectively. We used CI5 data on numbers of cases and person-years from the YT and nationwide Canadian registries, available by cancer site, sex, and 5-year age group. CI5X includes disaggregated data by gastric cancer subsite; however, at the time of analysis, CI5XI did not include stratification within subsite groupings, precluding the identification of subsite-specific cases by age group and sex. Furthermore, these data sources do not contain histological information that would permit estimates based on the Lauren histological classification of gastric cancers [46]. For NT-specific analyses, we used data provided directly by the NT government. These data included gastric cancer cases by subsite, sex, 5-year age group, Indigenous status, and year of diagnosis during 1997 and 2015. The NT dataset included dichotomized Indigenous status linked from vital statistics data housed by the NT government and mid-year populations for estimating person-years.

### 4.3. Statistical Analyses

We aimed to describe the incidence of gastric cancer overall and by subsite across study groups in the NT and YT and to compare these estimates to those observed in other populations. Our analyses used an estimation approach: We estimated relevant parameters to describe their magnitude and used confidence intervals (CIs) as measures of precision, to report the range of values compatible with the expected degree of random error in the data [47,48]. Accordingly, our comparisons of rates across populations were not based on statistical significance testing; instead, we focused on the magnitude, direction and precision of estimated associations.

We estimated the proportional incidence of NCGC as the number of NCGC cases divided by the total number of gastric cancer cases; to report the precision of estimated proportional incidence, we present 95% confidence intervals (CIs) based on the binomial exact distribution. To permit comparisons across groups with varying population age distributions, we estimated a weighted average of age-specific incidence rates using weights from a standard population, a process known as standardization. Here, we used two methods to accomplish this: standardized incidence ratios (SIRs) and age-standardized cancer incidence rates (ASRs).

SIRs are the functional equivalent of age- and sex-standardized incidence rate ratios using the study population distribution as the standard. We estimated SIRs by sex for all gastric cancer combined, CGC, and NCGC in YT (2003–2012) and NT Indigenous and non-Indigenous populations (1997–2015); to report the precision of estimated SIRs, we present 95% CIs based on the Poisson distribution. As the referent, which is the denominator, we used incidence rates in Canada; these were restricted to the years 2003–2007 due to the limited availability of data by gastric cancer subsite, age group, and sex. For SIRs, the numerator of the ratio is the total number of cases in the study group, which does not require stratification of cases by age group; thus, for SIRs we were able to include cases from a longer time period for YT despite the shorter time period for which the required data were available for Canada. SIRs provide a comparison of rates in the numerator and denominator populations unconfounded by the variables used for standardization; however, one SIR cannot be compared to another if the numerator populations differ substantially on the distribution of these variables.

For global comparisons, we estimated age-standardized cancer incidence rates (ASRs) by gastric cancer subsite, geography, and sex; to report the precision of estimated ASRs, we present 95% CIs based on the Poisson distribution. Due to limitations in the years of data available by subsite that were also disaggregated by age group and sex, ASRs by gastric cancer subsite for Canada and YT were limited to 2003–2007; ASRs for NT included data from 1997–2015. ASRs were age-standardized using the World Standard Population as defined by Segi [49]; we used world and regional CGC and NCGC ASR estimates reported by Colquhoun, et al. (2014) [2]. All data analyses, including SIR and ASR estimates and their CIs, were performed using SAS 9.4 software as outlined in the SAS/STAT^®^ 14.3 User’s Guide [50].

### 4.4. Case Definitions and Sensitivity Analyses

In the CI5 and NT datasets, gastric cancer cases were classified according to the *International Classification of Diseases, 10th Revision* (ICD-10): Gastric cancer (C16), CGC (C16.0), and NCGC (C16.1-9). NCGC typically includes the following anatomical regions of the stomach: fundus, body, antrum, pylorus, lesser curvature, and greater curvature (C16.1-6) [19,32,33]. However, the CI5X datasets group cancers of the greater and lesser curvature (C16.5 and C16.6) with cancers classified as overlapping and not-otherwise specified (C16.8 and C16.9). Knowing that a portion of this group includes NCGC cases (C16.5-6), and that other works have allocated some or all C16.8-9 sites to NCGC when data were limited [2,34,35,36], here we chose to classify C16.5-9 as NCGC. Because some of these cases might have been misclassified CGC cases, we also estimated SIRs and ASRs for NCGC defined as C16.1-4. Furthermore, in addition to age-standardization using the world population, we also repeated analyses using the Canadian 2011 standard population to permit comparisons across populations within Canada (Appendix A).

## 5. Conclusions

This study has, for the first time, estimated the incidence of CGC and NCGC in NT and YT populations. Consistent with other work describing gastric cancer among northern and Indigenous populations, we found that NT and YT populations were disproportionately impacted by gastric cancer, particularly of the non-cardia gastric subsite. This was especially true for Indigenous populations: NCGC incidence rates among NT Indigenous men were 2.7 times the rates among all men in Canada, while rates among NT Indigenous women were 3.1 times the rates among all women in Canada. Similarly, age-standardized rates of NCGC among Indigenous NT residents were comparable to global regions where there is a high burden of NCGC. These estimates provide new insights into the burden of gastric cancer in northern Canadian populations; as a result, this work has value for informing gastric cancer control strategies aimed at reducing the disproportionate gastric cancer disease burden of these populations.

## Figures and Tables

**Figure 1 cancers-11-00534-f001:**
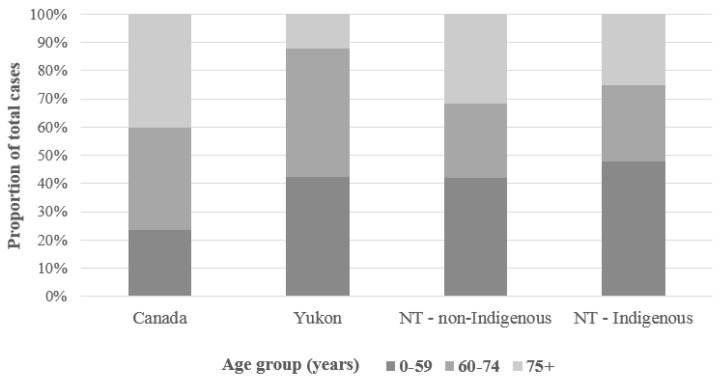
Age distribution of total gastric cancer cases by population; northern territories, Canada; various years. Canada and Yukon include cases diagnosed 2003–2012; Northwest Territories (NT) includes cases diagnosed 1997–2015.

**Figure 2 cancers-11-00534-f002:**
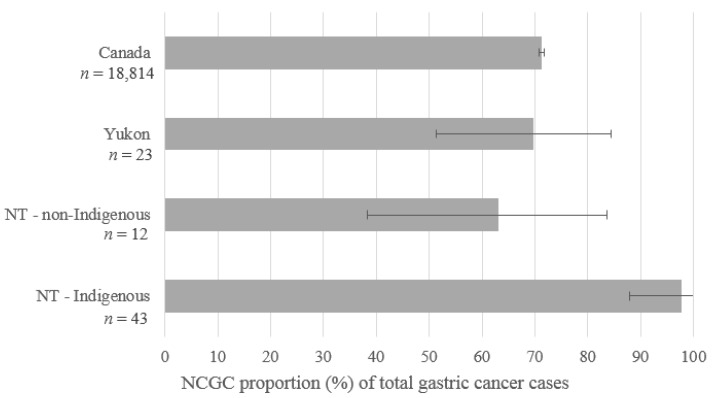
Proportional NCGC incidence (NCGC cases divided by total gastric cancer cases); northern territories, Canada; various years. Canada and Yukon include cases diagnosed 2003–2012; Northwest Territories (NT) includes cases diagnosed 1997–2015. Non-cardia gastric (NCGC) defined as C16.1-9. 95% confidence intervals are based on the binomial exact distribution.

**Figure 3 cancers-11-00534-f003:**
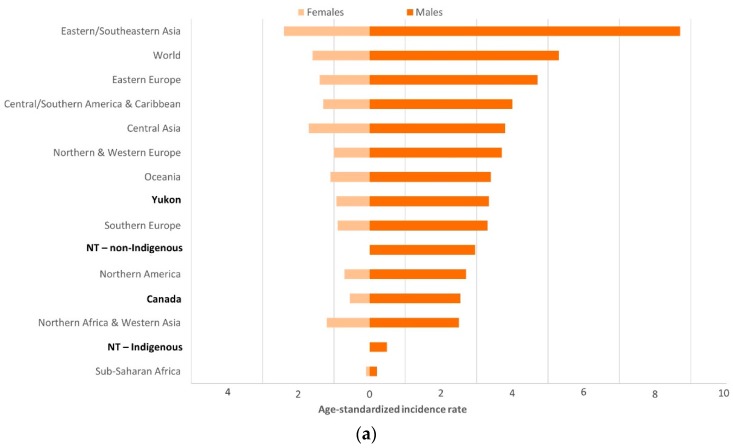
(**a**) Estimated cardia gastric cancer age-standardized incidence rates (ASRs per 100,000) by population and sex; (**b**) Estimated non-cardia gastric cancer ASRs (per 100,000) by population and sex; northern territories, Canada, and world regions, various years. Age-standardized incidence rates (ASRs) standardized to the World Population. Canada and Yukon include cases diagnosed during 2003–2007; Northwest Territories (NT) includes cases diagnosed during 1997–2015. World and regional cardia and non-cardia gastric cancer ASR estimates from Colquhoun, et al. (2014) [2].

**Table 1 cancers-11-00534-t001:** Number of gastric cancer cases and person-years by gastric cancer subsite, population and sex; northern territories, Canada; various years.

Population	Gastric Cancer Subsite ^1^	Calendar Years	Men		Women	
			Number of Cases	Person-Years	Number of Cases	Person-Years
Canada	All	2003–2012	16,872	144,495,378	9510	147,171,135
	CGC	2003–2007	3151	79,864,738	857	81,320,815
	NCGC	2003–2007	6489	79,864,738	4544	81,320,815
Yukon	All	2003–2012	21	169,680	12	162,500
	CGC	2003–2012	7	169,680	3	162,500
	NCGC	2003–2012	14	169,680	9	162,500
NT—non-Indigenous	All	1997–2015	16	225,396	3	199,893
	CGC	1997–2015	7	225,396	0	199,893
	NCGC	1997–2015	9	225,396	3	199,893
NT—Indigenous	All	1997–2015	26	215,886	18	214,022
	CGC	1997–2015	1	215,886	0	214,022
	NCGC	1997–2015	25	215,886	18	214,022

^1^ CGC, cardia gastric cancer; NCGC, non-cardia gastric cancer. CGC defined as C16.0; non-cardia gastric NCGC defined as C16.1-9; NT, Northwest Territories.

**Table 2 cancers-11-00534-t002:** Number of cases and estimated standardized incidence ratios (SIR) comparing northern populations to Canada as a whole on gastric cancer incidence rates by gastric cancer subsite, population, and sex; various years.

Population	Gastric Cancer Subsite ^1^	Men				Women			
		Cases ^2^	SIR ^3^	95% CI ^4^		Cases ^2^	SIR ^3^	95% CI ^4^	
Yukon	All	21	1.3	0.78	1.9	12	1.8	0.89	3.0
	CGC	7	1.2	0.48	2.5	3	2.6	0.53	7.5
	NCGC	14	1.3	0.71	2.2	9	1.5	0.71	2.9
NT— non-Indigenous	All	16	1.1	0.61	1.7	3	0.52	0.11	1.5
	CGC	7	1.3	0.51	2.6	0	0.00	0.00	3.8
	NCGC	9	0.94	0.43	1.8	3	0.62	0.13	1.8
NT—Indigenous	All	26	1.9	1.2	2.7	18	2.6	1.5	4.1
	CGC	1	0.21	0.01	1.2	0	0.00	0.00	3.3
	NCGC	25	2.7	1.7	3.9	18	3.1	1.8	4.8

SIR, standardized incidence ratio; CI, confidence interval; CGC, cardia gastric cancer; NCGC, non-cardia gastric cancer; NT, Northwest Territories. ^1^ CGC defined as C16.0; NCGC defined as C16.1-9; ^2^ Yukon includes cases diagnosed during 2003–2012; NT includes cases diagnosed during 1997–2015; ^3^ SIRs compare study population to Canada during 2003–2007; ^4^ 95% CIs are based on the Poisson distribution.

**Table 3 cancers-11-00534-t003:** Estimated gastric cancer age-standardized incidence rates (ASRs per 100,000) by subsite, population, and sex; northern territories and Canada; various years.

Population	Gastric Cancer Subsite ^1^	Men				Women			
		Cases ^2^	ASR ^3^	95% CI ^4^		Cases ^2^	ASR ^3^	95% CI ^4^	
Canada	All	16,872	7.0	6.9	7.1	9510	3.2	3.1	3.3
	CGC	3151	2.5	2.4	2.6	857	0.6	0.6	0.6
	NCGC	6489	5.0	4.9	5.1	4544	2.8	2.7	2.9
Yukon	All	21	9.2	5.2	13.2	12	5.8	2.4	9.2
	CGC	3	3.3	0.0	7.2	1	0.9	0.0	2.7
	NCGC	6	6.1	1.1	11.1	4	4.5	0.0	9.1
NT—non-Indigenous	All	16	8.8	3.9	13.7	3	2.0	0.0	4.3
	CGC	7	3.0	0.6	5.4	0	0.0	0.0	0.0
	NCGC	9	5.8	1.5	10.1	3	2.0	0.0	4.3
NT—Indigenous	All	26	13.8	8.4	19.2	18	7.7	4.1	11.2
	CGC	1	0.5	0.0	1.4	0	0.0	0.0	0.0
	NCGC	25	13.3	8.0	18.6	18	7.7	4.1	11.3

ASR, age-standardized incidence rate; CI, confidence interval; CGC, cardia gastric cancer; NCGC, non-cardia gastric cancer; NT, Northwest Territories; ^1^ CGC defined as C16.0; NCGC defined as C16.1-9; ^2^ For Canada and Yukon, all gastric cancer cases combined include cases diagnosed during 2003–2012; gastric cancer subsites include cases diagnosed during 2003–2007. NT includes cases diagnosed during 1997–2015; ^3^ ASRs standardized to the World Population; ^4^ 95% CIs are based on the Poisson distribution.

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
