# Peer review of "Gastric Cancer in Northern Canadian Populations: A Focus on Cardia and Non-Cardia Subsites"

_cancers, 2019, doi:10.3390/cancers11040534_

Reviewer 1 Report

The study estimated epidemiological difference in cardias and non cardias gastric cancer in different population of Canadian country.

comments:

due to the importance of H. pylori in resident and endogenous population the title of IgG against H. pylori or an alternative test must be added in the study. A distinguished between intestinal and diffuse ore between proximal (exclded cardias) and distal cancer could be added in the type of non cardia gastric cancer.

introduction can be improved by change of epidemiological change occurring into the world ...in particular regarding increase in women prevalence and genetic/hereditary importance in endogenous incidence of gastric cancer

Author Response

Response to Reviewer 1

due to the importance of H. pylori in resident and endogenous population the title of IgG against H. pylori or an alternative test must be added in the study.

We are uncertain about the meaning of this statement. In particular, it is unclear what the reviewer means by adding a diagnostic test to the study. Here, we aim to describe the burden of the gastric cancer topographical subsites, cardia and non-cardia, by estimating incidence rates using cancer registry data. Given this goal, the rationale for adding another variable is not clear. If the reviewer is suggesting that we convert this into an ecologic study of the association between H. pylori prevalence and gastric cancer rates, the size of the population subgroups included in our study would make this a methodologically unsound exercise. Moreover, accurate estimates of H. pylori prevalence are not available for all of the populations included in our study: surveillance data on H. pylori testing is not routinely collected. This applies to all H. pylori tests, including testing for IgG antibodies. 

A distinguished between intestinal and diffuse ore between proximal (exclded cardias) and distal cancer could be added in the type of non cardia gastric cancer.

Here, we used information publicly available through the International Agency for Research on Cancer, which does not contain detailed information on the histological features of gastric cancer cases. To clarify this, we have added a sentence in the Materials and Methods section of the manuscript (lines 514-515). Furthermore, we identified cardia and non-cardia gastric cancers using case definitions previously used (see lines 566-569). A further disaggregation of non-cardia gastric cancer subtypes might distinguish non-cardia cancers of the proximal region (such as the fundus, C16.1) from more distal areas of the stomach (such as the pylorus, C16.5).  However, given that there were fewer than 10 total cases of non-cardia gastric cancer in half of the study groups during the study period, the authors do not believe that further disaggregation would offer meaningful benefit.

introduction can be improved by change of epidemiological change occurring into the world ...in particular regarding increase in women prevalence and genetic/hereditary importance in endogenous incidence of gastric cancer 

Our Introduction aims to provide the reader with sufficient background to contextualize the purpose, methods, and results of our analyses. We summarize literature on sex and geographic variation. We also describe the epidemiology of gastric cancer for populations relevant to our analyses: namely differences that exist between ethnic groups such Indigenous populations and their non-Indigenous counterparts. Because our analyses do not focus on time trends, our description of changes in gastric cancer incidence over time and the reason for these changes in the Introduction is brief. We do not believe that reversals in falling incidence and male predominance among non-Hispanic white populations in the US are necessary to report in the Introduction. Instead, where relevant, we reference more detailed work in the Discussion. In particular, we note that NCGC rates may be increasing in younger non-Hispanic white populations in the United States (lines 476-479). To highlight this trend in women, we have added, “…particularly among women” in line 479.

Thank you for the suggestion to reference genetic risk factors in the Introduction. We have added a statement about the potential contribution of these factors to differences in gastric cancer incidence across geographies in lines 47-49 and lines 55-56.

Reviewer 2 Report

The paper discusses NCGC and CGC standardized incidence ratios within Canada among indigenous and non-indigenous populations, and compare them to other regions in the world using age-standardized rates.
The section Material and Method should be before the section Results. The paper seems very confusing in the current presentation.
- Results
Figure 2. Please change the x-axis label to NCGC proportion of total cases
L121-122: This sentence seems incomplete. Were they apparent higher rates? Because the confidence interval includes 1.

L124-127: Statistically imprecise means high variance or large confidence intervals. Except for Yukon women, CGC SIR are statistically less imprecise than NCGC SIR in NT - Indigenous men and Yukon men. 

For example, the 95% CI for NCGC SIR for NT indigenous women goes from 1.7 to 4.5, while it goes from 0.31 to 2.09 for Yukon men. Moreover, SIR for Yukon men and NT non-indigenous men are close to 1. Therefore, statistically imprecise estimates do not justify that the data does not agree with the authors' hypothesis for these populations.

The estimator be SIR could be biased when the number of events is small. If so, then the authors could justify the lack of agreement with their hypothesis. Notice that the sample size is the size of each population, which is large. The event is rare. The statement should be rewritten.
On the other hand, the authors also stated in L232-240 that SIR was used to compare the number of cases, which takes into account the small number of events. This statement says that SIR is an unbiased estimator even when the number of events is small, which is a contradiction of the previous statement.

Author Response

Response to Reviewer 2

 The section Material and Method should be before the section Results. The paper seems very confusing in the current presentation.

We followed the format required by the Journal. If the Journal is willing to modify the order of reported sections, we would be happy to move the Material and Methods before the Results.

 Figure 2. Please change the x-axis label to NCGC proportion of total cases

Thank you for the suggestion. We have re-labeled the x-axis of Figure 2 so that it reads, “NCGC proportion of total gastric cancer cases” and we edited the legend to begin with “Proportional NCGC incidence (NCGC cases divided by total gastric cancer cases)” to improve clarity.

 L121-122: This sentence seems incomplete. Were they apparent higher rates? Because the confidence interval includes 1.

Our analysis uses an effect estimation approach rather than statistical significance testing, as recommended by contemporary authoritative epidemiological methods texts such as Rothman et al (2008) and Rothman (2012). Using this approach, the estimated magnitude and direction of associations are assessed separately from the statistical precision, and confidence intervals (CI) are interpreted as the range of values compatible with the data rather than as indicating the existence of an effect in a particular direction or of a particular magnitude if and only if the CI excludes 1. We have added a note about this within the Materials and Methods section of the manuscript (lines 523-527). We have also moved a sentence describing the precision of estimates later in the paragraph being referenced here to immediately after lines 121-122 to improve clarity.

Rothman, K. J., Greenland, S. & Lash, T.L. (2008). Precision and Statistics in Epidemiologic Studies. In Modern Epidemiology (3rd ed., pp. 151-152). Philadelphia, PA: Wolters Kluwer.

Rothman, K. J. (2012). Random Error and the Role of Statistics. In Epidemiology: An Introduction (2nd ed., pp.151-152). New York, NY: Oxford University Press.

L124-127: Statistically imprecise means high variance or large confidence intervals. Except for Yukon women, CGC SIR are statistically less imprecise than NCGC SIR in NT - Indigenous men and Yukon men. For example, the 95% CI for NCGC SIR for NT indigenous women goes from 1.7 to 4.5, while it goes from 0.31 to 2.09 for Yukon men. Moreover, SIR for Yukon men and NT non-indigenous men are close to 1.

We assess precision based on the span of values included in the confidence interval (CI). Because ratios are on a relative (multiplicative) scale, appropriate comparison of the precision of their confidence intervals is based on how many times greater the upper limit is than the lower limit; this can be calculated by dividing the upper confidence limit by the lower confidence limit (UL/LL). For NT Indigenous men, the CI width that was originally reported (1.6-3.7) is 2.3-fold for NCGC, much more precise than the indeterminate distance between the confidence limits for CGC (0-0.62) given that this CI spans values from<2 to="">100 times smaller than 1.0; for Yukon men, the NCGC CI (0.62-2.0) clearly has a narrower span than the CGC CI (0.31-2.09). The authors have updated these CIs so that they are based on a Poisson distribution; however, their interpretation remains unchanged. The distance from 1 of SIRs has no bearing on the precision indicated by the CIs.

Therefore, statistically imprecise estimates do not justify that the data does not agree with the authors' hypothesis for these populations.

The reviewer's meaning here is unclear. This analysis is entirely descriptive. It does not aim to test hypotheses and, therefore, we have not stated any. It is not a goal of our analysis to draw inferences about agreement between the data and hypotheses. Our goal is to estimate rates and SIRs and to accurately describe the statistical precision of our estimates.  

The estimator be SIR could be biased when the number of events is small.

We assume the reviewer means that the SIR could be inaccurate when the number of events is small, given that the main concern about small numbers is the introduction of random error while bias refers to non-random error (see Rothman et al (2008) and Rothman (2012)).

Rothman, K. J., Greenland, S. & Lash, T.L. (2008). Modern Epidemiology (3rd ed). Philadelphia, PA: Wolters Kluwer.

Rothman, K. J. (2012). Epidemiology: An Introduction (2nd ed.). New York, NY: Oxford University Press.

If so, then the authors could justify the lack of agreement with their hypothesis.

Please note our response above about descriptive analyses versus hypothesis testing.

Notice that the sample size is the size of each population, which is large. The event is rare. The statement should be rewritten. 
The reviewer’s point about large populations and rare events is unclear.

 On the other hand, the authors also stated in L232-240 that SIR was used to compare the number of cases, which takes into account the small number of events. This statement says that SIR is an unbiased estimator even when the number of events is small, which is a contradiction of the previous statement.

We are uncertain about which previous statement is being referenced in the last sentence above. If the reviewer is describing the first sentence of the paragraph on lines 232-233 (currently 463-465), the reviewer may have misread ‘incidence rate estimates’ as SIRs. Here, we refer to a limitation of age-standardized incidence rates (ASRs) and then outline how the use of SIRs may be beneficial, citing Szklo and Nieto (2014). To minimize potential misinterpretation, we have replaced “incidence rate” with “ASR” in the first sentence of that paragraph (line 464).

 Szklo, M. & Nieto, J (2014). Epidemiology: Beyond the Basics (3rd ed.). Burlington, MA: Jones & Bartlett Learning.

 Round  2

Reviewer 1 Report

The responses provided are suffcient